# A community-based vector control intervention "Slash and Clear" implemented in two onchocerciasis-endemic foci in South Sudan

**Thomson Luroni Lakwo[1], Joseph Siewe Fodjo[2], Stephen Raimon Jada[3], Peter Alinda[1], Moses Tionga[2], Constantino Doggale Remijo Marcello[4], Deng Gai Dual War[4], Marina Saleeb[2], Robert Colebunders** [2,5]*

**1** Ministry of Health, Vector Control Division, Kampala, Uganda, **2** Global Health Institute, University of Antwerp, Antwerp, Belgium, **3** Amref Health Africa, Juba, South Sudan, **4** Ministry of Health, National Malaria Control Program, Juba, South Sudan, **5** Department of Tropical Disease Biology, Liverpool School of Tropical Medicine, Liverpool, United Kingdom

* robert.colebunders@uantwerpen.be

## Abstract

### Background

Despite several rounds of ivermectin treatment, onchocerciasis transmission persists in Mvolo and Mundri West Counties (Western Equatoria State, South Sudan). A community-based "Slash and clear" (S&C) vector control method was recently introduced, but its effectiveness remains unclear.

### Methods

Between October 2023 and November 2024, entomological studies were conducted to map blackfly breeding sites along River Naam (Mvolo County) and River Yei (Mundri West County). Following baseline assessment of daily biting rates at multiple catching sites, one round of S&C was implemented at selected intervention sites. Monthly biting rates (MBR) were monitored over a follow-up period of 8–14 months and compared between intervention and control sites.

### Results

Thirteen breeding sites of *Simulium damnosum, sensu lato*. were found on the Naam and Yei Rivers. Biting rates were consistently higher during the rainy season than the dry season across both Counties. On the Naam River (14 months of follow-up), Mann-Kendall trend tests showed non-significant reductions in MBRs at both intervention (tau = −0.038, p-value = 0.881) and control sites (tau = −0.135, p-value = 0.313). Similar non-significant changes were observed on the Yei River (8 months follow-up), with tau = 0.046 (p-value = 0.820) and tau = 0.163 (p-value = 0.363) for intervention and control sites, respectively. Generalized additive models (GAM)

**Data availability statement:** The authors confirm that all data underlying the findings are fully available without restriction. All relevant data are within the paper and its Supporting Information files.

**Funding:** This work was supported by the Research for Health in Humanitarian Crises (R2HC) programme (Project ID: 78719) to Amref Health Africa, South Sudan and the Research Foundation Flandres (FWO) (grant G0A0522N) to RC. SRJ received a salary from R2HC. The funders had no role in study design, data collection and analysis, decision to publish, or preparation of the manuscript.

**Competing interests:** The authors have declared that no competing interests exist.

regression analysis indicated that seasonality was the only significant predictor of MBR, with increased biting rates during the rainy season ($p < 0.001$). A single round of S&C at baseline did not result in significant reductions in MBRs (p-value = 0.651 in Mvolo and p-value = 0.531 in Mundri West).

## Conclusion

Blackfly biting rates in Mvolo and Mundri West Counties are strongly influenced by seasonal variations, peaking during the rainy season. Our findings indicate that a single round of S&C is insufficient to reduce blackfly biting in the medium term. Repeated and strategically timed annual implementation of S&C is likely required to achieve significant and lasting vector control impacts.

## Author summary

Onchocerciasis, or river blindness, is a parasitic disease transmitted through blackfly bites. The primary control strategy has been the mass distribution of ivermectin, but this approach has not consistently achieved the desired outcomes in many endemic communities. Recently, an alternative method known as "slash and clear" (S&C) has shown potential in reducing transmission. This technique involves removing riverside vegetation where blackflies typically breed, thereby reducing their population. To test the effectiveness of S&C in South Sudan, researchers mapped blackflies breeding sites along the Naam River in Mvolo County and the Yei River in Mundri West County. They measured how often people were bitten before and after the S&C intervention. The results showed that on both rivers, the effects of a single round of S&C were not sustained during the subsequent months especially in the rainy season when blackfly abundance tends to increase. This indicates that S&C might need to be repeated several times each year to achieve a more substantial and lasting reduction in blackfly biting rates.

## Introduction

Onchocerciasis is still a public health problem at global level with 99% of the cases registered in sub-Saharan Africa. *Onchocerca volvulus*, the causative parasite of onchocerciasis, is transmitted to man via infective bites of the blackfly insect (*Simulium*) acting as the vector. Onchocerciasis is prevalent in approximately half of South Sudan [1]. One of the most affected states of South Sudan is Western Equatoria with onchocerciasis hotspots in Maridi, Mvolo, and Mundri West Counties [2–4].

Since the 1990s, global efforts to control and eliminate onchocerciasis have relied on large-scale preventive chemotherapy through mass drug administration (MDA) of ivermectin, which kills the larval stages (microfilariae) [1]. In some areas, MDA has been complemented by vector control strategies, such as aerial larviciding, to reduce

blackfly populations [1]. Together, these approaches form the basis of onchocerciasis control and elimination programmes. Vector control is a critical adjunct to ivermectin MDA for achieving onchocerciasis elimination, particularly in settings where: vector densities are high, transmission persists despite MDA, and especially in geographically isolated foci where it has proven highly effective, such as in in the Victoria Nile [5] and Mount Elgon [6] foci in Uganda and Bioko Island in Equatorial Guinea [7]. However, in South Sudan where are onchocerciasis foci are not geographically isolated and rivers may contain continuous breeding sites and where resources are limited, implementing vector control is challenging.

In 2017, a high prevalence of onchocerciasis-associated epilepsy (OAE) and high ongoing *O. volvulus* transmission was documented in Maridi [8]. This high prevalence was found to be related to the high ongoing *O. volvulus* transmission at the Maridi Dam, the only blackfly breeding site in the area. Therefore, in 2019, in an effort to supplement ivermectin treatment, a low-cost community-based vector control method, "Slash and Clear" (S&C), was implemented at the Maridi dam, the local blackfly breeding site [9]. It took only four volunteers and four working days to remove the vegetation at the Maridi dam spillway and reduce the blackfly biting rate by more than 50% [10]. Based on these promising results, the NTD program in collaboration with Amref Health Africa, South Sudan wanted to implement this strategy in Mvolo and Mundri West, counties with also a high OAE prevalence. Prior to the implementation of this intervention an entomological and qualitative study was conducted to find blackfly breeding sites, and explore the feasibility, and acceptability of a S&C intervention by the local communities.

Entomological findings and the initiation of the S&C in Mvolo have been described previously [11]. In this paper, we report the one-year follow-up results of the S&C intervention in Mvolo and the entomological findings and initiation of the S&C in Mundri West County.

## Materials and methods

### Ethics statement

This study obtained ethical approval from the Ministry of Health of South Sudan (MOH/RERB/P35//15/05/2023-MOH/RERB/A/35/2023) and the University of Antwerp, Belgium (BUN B3002023000045). The study was conducted in conformity to the Declaration of Helsinki and the South Sudan Ministry of Health Ethical guidelines regarding handling of human subjects. There were community engagements before commencement of the slash and clear and human landing catches activities in Mvolo and Mundri West Counties. Formal verbal informed consent was obtained from the participants in these activities.

### Description of study areas

The Western Equatoria State where this study was conducted lies in the South Western region of South Sudan, on the border with the Democratic Republic of Congo. The area experiences two main seasons, namely the rainy season from April to October (rainfall levels > 100mm) and the dry season from November to March (rainfall levels < 100mm) [9]. Within the Western Equatoria State, the two counties of interest in this study were the Mvolo County and the Mundri West County.

### Mvolo county

This County is primarily a wooded grassland with the River Naam being its main watercourse. It meanders along its course, occasionally splitting into two or three sub-tributaries [11]. Breeding of *S. damnosum* in Mvolo is confined to a very short stretch measuring approximately 3–5 km, located near human settlements [11] (Fig 1). A S&C community-based vector control intervention was implemented in October 2023 for two days by six volunteers at Dogoyabolu, also commonly called "Kpatiborokgba" (Latitude: 6.05341; Longitude: 29.94885) on the River Naam [11]. This was the only suitable site for this strategy despite the predominant rock outcrops in the middle of the river and along its banks [11]. Two control sites (site 1: 6.04698; 29.95107, site 2: 6.05489; 29.9462) were chosen not very far from the intervention site [11]. The

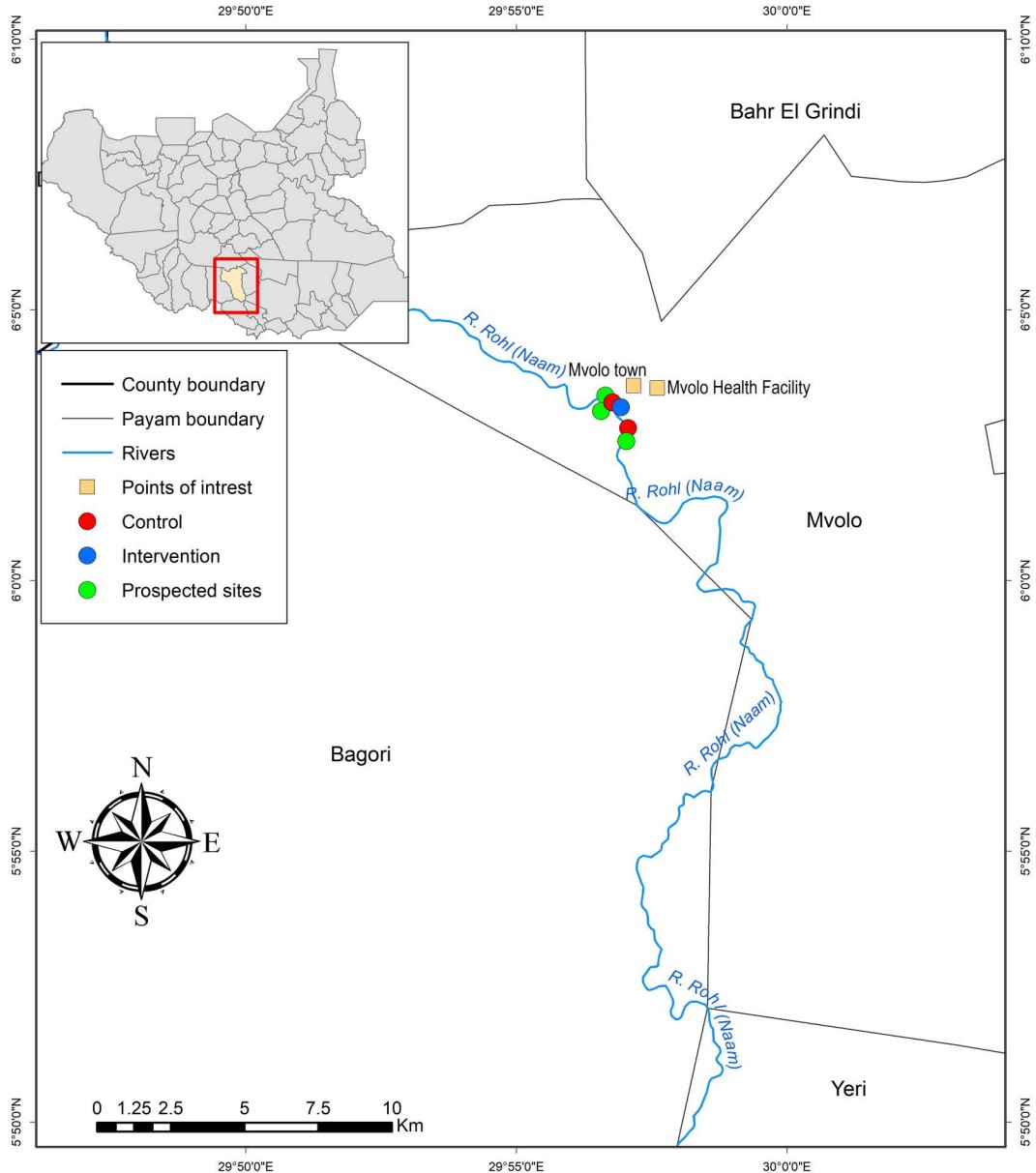

**Fig 1. Blackfly breeding sites (one intervention and two control sites) along River Naam in Mvolo County, Western Equatoria, South Sudan.** Figure was made using Humanitarian Data Exchange, https://data.humdata.org/.

Domilara and Main Bridge control sites were located 1.5 km and 2.5 km, respectively, from the intervention site. The short distances between the intervention and control sites were due to the limited stretch of *Simulium damnosum* breeding habitat along the River Naam in Mvolo.

## Mundri west county

Mundri West County borders Mvolo County to the north, Mundri East County to the east, Maridi County to the west and the Central Equatoria State (Yei, Lainya and Juba Counties) to the south. It is a savanna with 97% forest cover and 2.2%

grassland [12], predominantly featuring tall-growing grass hyparrhenia rufa grass (Jaragua). The main river in this County is the River Yei which is located about 2km from Mundri town (Fig 2). Amadi Payam is an administrative division within Mundri West County, predominantly inhabited by the Moru and Avukaya ethnic groups. It hosts local government offices, including those of the chief and other local administrators. Permission to conduct activities in the community is always obtained through these offices..

The River Yei is one of the largest rivers in South Sudan with its source in Panyana Village, Lujule payam in Morobo County of Central Equatoria. The name Yei was derived from the word "yii" in Kaliko language, meaning "water" [13]. The

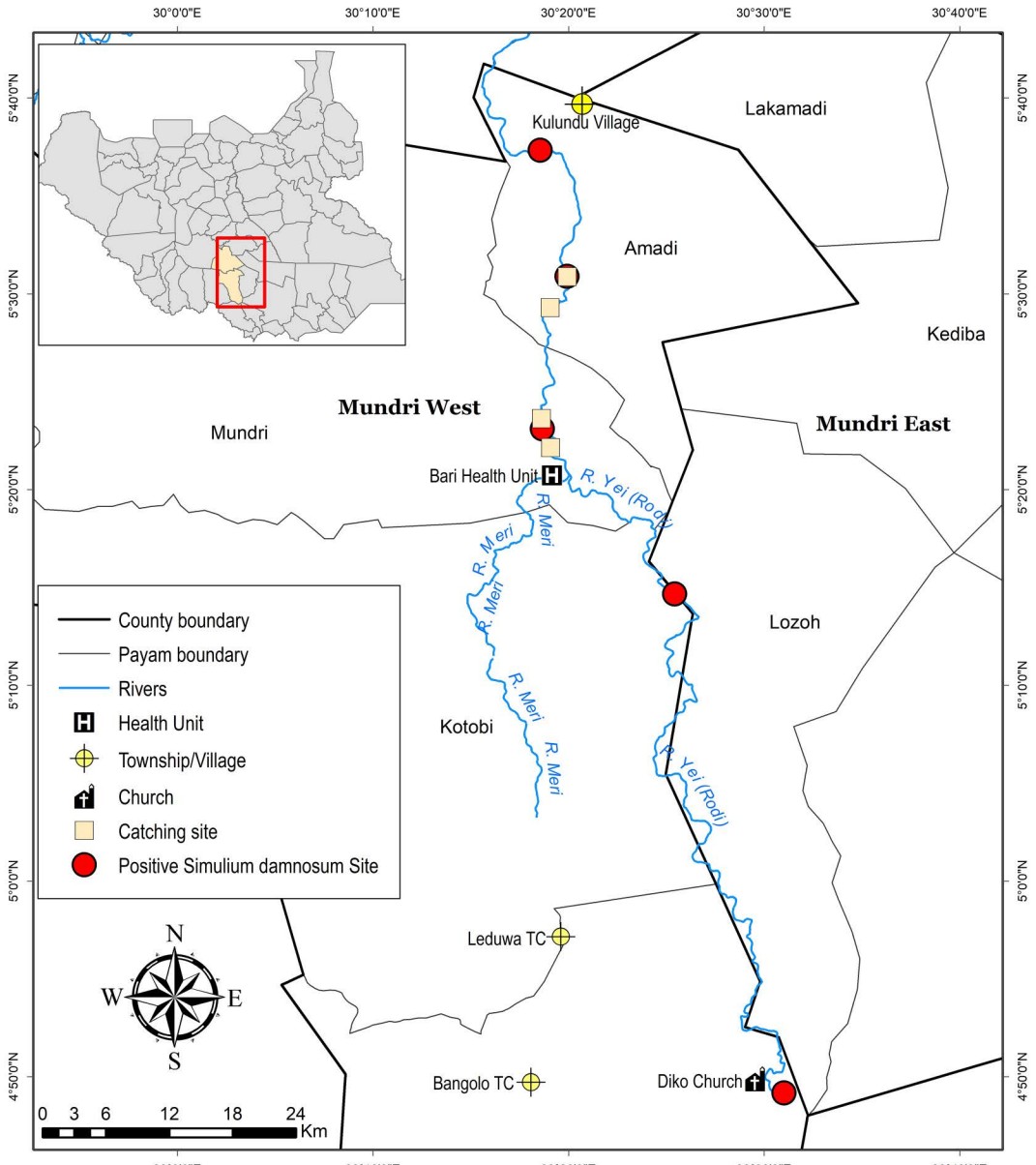

**Fig 2. Location of some *S. damnosum* breeding sites and four established catching sites (two intervention & 2 control sites) along R.** Yei in Mundri West County, Western Equatoria, South Sudan. Figure was made using Humanitarian Data Exchange, https://data.humdata.org/.

River Yei is categorized as a large river because most of its stretches measure between 100 to 250m in width. In some sections, it splits into sub-tributaries, and in areas with rock outcrops it creates fast flowing water, which serve as potential breeding sites for blackflies. There are many rapids along River Yei which are well known by the local communities. Populations practice subsistence agriculture based on the cultivation of cassava, sorghum, groundnuts and vegetables along the River Yei, especially during the dry season. Fishing is also practiced, though it is less prominent compared to that along the River Naam in Mvolo.

The dry season weather in Mundri West is characterized by very high temperatures and low relative humidity. Temperatures range from 26°C to 42°C, starting at around 26°C in the early morning (6:00–8:00 AM), rising to a peak of 42°C in the early afternoon (2:00 PM), and gradually decreasing toward the evening (5:00–6:00 PM) [14]. Mundri West is also one of the counties in South Sudan affected by the warmer than usual temperature anomalies [15].

## Study design

A prospective design was adopted, involving repeated measurements of biting rates over time. Rivers were purposively selected based on known onchocerciasis-endemicity in the respective counties. After identifying suitable study areas, intervention sites (for S&C implementation) and control sites were established.

## Study procedures

**Mapping of blackfly breeding sites.** The two rivers (Naam and Yei) were surveyed by experienced technical teams, including entomologists from South Sudan and Uganda. The team was accompanied by selected community members with extensive knowledge of the rivers and their tributaries. The primary aim of these surveys was to identify blackfly breeding sites and evaluate the feasibility of implementing S&C.

Due to the absence of detailed topographical maps, site selection relied on knowledge from nearby communities. The sites were accessed by vehicle or on foot depending on seasonal conditions—during the rainy season for the River Naam and during the dry season for the River Yei. At each visited site, collections of immature stages of *S. damnosum,* s.l. were conducted following established protocols [16]. Larval density at breeding sites was determined semi-quantitatively (<10 larva/pupae, 11–50 larvae/pupa, >50 larvae/pupae). For microscopic identification, larvae and pupae were harvested from vegetation and placed in petri dishes with a small drop of water. Larvae were examined for characteristic dorsal scales while pupae were dissected to expose their gill filaments and identified using WHO-documented keys [16]. No attempt was made to collect or identify larvae from species other than *S. damnosum.*

**Collection of adult biting *Simulium.*** To measure human-biting rates of blackflies the human landing catch (HLC) method was chosen, which is still considered by the World Health Organisation (WHO) as the gold standard method for this purpose, Esperanza Window traps are an alternative to collect blackflies. While Esperanza traps offer ethical and operational advantages by reducing human exposure to blackfly bites, they currently face technical limitations: They measure attraction rather than true biting rates, show variable correlation with HLC data, and are not yet validated as a reliable substitute.

Volunteers, residing near the Naam and Yei Rivers were selected based on recommendations from local chiefs. The research team trained these volunteers in basic blackfly collection and preservation techniques [11] and all were treated with ivermectin. A total of 14 vector collectors were trained: eight were deployed at four catching sites along the River Yei, and six were assigned to three catching sites along the Naam River. At each site, two collectors were operated within 500 meters of known blackfly breeding sites.

Collections were performed by two operators per site working in alternating shifts from 06:00–17:00 h [16]. Collection times were adjusted to South Sudan standard time. The collectors sat at the river banks and exposed the lower part of their legs. Any blackfly attempting to land for a blood meal was captured before feeding by inverting a small plastic tube over it, after which the tube was immediately sealed with a cap. Tubes were pooled according to the hour of collection at

each site. Baseline data collection was consecutively done for 7–9 days, while monthly biting rates assessments lasted 4 consecutive days per month on both rivers.

### "Slash and clear" implementation and monitoring

The implementation of S&C along the Naam and Yei Rivers was guided by findings from the surveys in the two rivers. The research team engaged with community leaders, including village chiefs and community members in Mvolo and Amadi, to discuss the intervention. A total of 16 youths from Mvolo and Amadi were selected and trained to carry out S&C along the Naam and Yei Rivers, respectively. These individuals, primarily fishermen aged 20–35 years, were chosen for their swimming experience. Each team was equipped with machetes and necessary protective gear, including rubber boots.

At designated intervention sites, field teams used sharpened machetes to cut and remove trailing vegetation entangled on submerged rocks, discarding the cut vegetation along the riverbank. Slashing was conducted 700 meters upstream and 200 meters downstream from the target sites, adhering to procedures previously described [17]. Due to different sizes of the intervention sites, slashing along the River Naam took three consecutive days (mid-October 2023), while clearing intervention site along the River Yei required seven consecutive days (March 2024).

### Follow-up biting rates assessments

Following the completion of S&C, blackfly population monitoring resumed immediately both at intervention and control sites. Vector collectors were supervised daily by site field supervisors and monthly by technical staff from the Ministry of Health and Amref Health Africa, Juba, South Sudan. All blackfly collection data were recorded on standardized collection forms [16], which were regularly checked for completeness and accuracy before being entered into an Excel database. Monitoring of biting rates was conducted monthly from end-October 2023 to November 2024 (14 months) along the River Naam, and from April 2024 to November 2024 (eight months) along the River Yei.

### Data handling and analysis

Blackfly collection forms were reviewed during site supervision by team members to ensure accuracy, completeness, and validity of the data. Monthly aggregated black fly data from each site were entered into a database and subsequently converted to Monthly Biting Rates (MBRs) as previously described [8]. Bites per person per day (BI/P/D) were calculated by dividing the number of blackflies caught by person-days worked.. Changes in MBR between baseline and in the month immediately following S&C was calculated for all sites, and expressed as percentage change relative to baseline MBR The following formula was used:

$$\% \text{ change} = [(\text{post intervention MBR} - \text{Baseline MBR})/\text{Baseline MBR}] \times 100$$

Since S&C was only implemented once at baseline in the intervention sites, its effect on MBR during the subsequent months could only be assessed by analyzing the entire follow-up period. Given the high variability of MBR across months, the overall trend of MBR evolution for each river was assessed using the Mann-Kendall trend test. The resulting z-statistic provided a measure of the trend as well as its direction (positive z for upward trend and negative z for downward trend), while the p-value indicated the statistical significance of the trend. Mann-Kendall test results were compared between intervention and control sites of the same river.

Furthermore, to account for the effect of season (dry vs rainy) on the biting rates, a generalized additive model (GAM) was set up to test the difference in MBR trends across seasons and study group (intervention vs control). For this model, log-transformed MBR (to which 0.001 was added to have only non-zero values) was used as outcome variable, while S&C intervention per site (yes vs no) and season (rainy vs dry) were used as fixed covariates. The output of the GAM will be reported in two parts: principal effects and smooth terms. The principal effects are the linear, average relationships

between predictors and outcome. The intercept is the average log-transformed biting rate in baseline conditions, and linear regression coefficients for each covariate are also provided. Smooth terms, on the other hand, explore the more complex non-linear trends over time (months of follow-up). The effective degrees of freedom (EDF) measure the smooth terms' complexity in the model. An EDF close to 1 means that the relationship is linear, and more than 1 implies a nonlinear trend with increasing flexibility. Zero EDF, however, means no significant trend was detected for that term [16].

For all analyses, a p-value less than 0.05 was considered as statistically significant. Data was analyzed using Microsoft Excel 2021 and R version 4.3.2.

## Results

### Mapping blackfly breeding sites

A total of 16 breeding sites were identified along the Naam and Yei Rivers (Table 1). Along the River Naam, two breeding sites were heavily infested (≥50 larvae/pupae) while the remaining sites had no detectable larvae or pupae, although adult flies were captured at some locations. Meanwhile along the River Yei, seven sites were heavily infested with *S. damnosum* larvae and pupae; two sites had light infestations (<10 larvae/pupae); while one site showed no presence of larvae or pupae. Overall, the River Yei had more sites with heavy infestations compared to the Naam River (Table 1).

### Collection of adult biting *Simulium*

At baseline, the bites per person per day (BI/P/D) along the River Naam ranged from 29.6 to 40.4 BI/P/D, with an overall BI/P/D of 36.1 bites per person per day (Table 2). Meanwhile, along the River Yei, the lowest BI/P/D was observed at the

**Table 1. Prospection of blackfly breeding sites on the Naam and Yei Rivers in Western Equatoria State.**

| Date | River | Site | GPS Location | | Altitude | Simulium damnosum | | |
|------|-------|------|--------------|--|----------|-------------------|--|--|
| | | | Latitude (degrees) | Longitude (degrees) | (meters) | Larvae | Pupae | Biting females |
| **NAAM RIVER SURVEYS-2023** | | | | | | | | |
| 6/10/23 | Naam | Dombolo | 6.06007 | 29.95279 | 465 | 0 | 0 | 0 |
| 6/10/23 | Naam | Domilara | 6.05489 | 29.9462 | 469 | 0 | 0 | + |
| 6/10/23 | Naam | Kpatiborokgba | 6.05341 | 29.94885 | 485 | +++ | +++ | + |
| 6/10/23 | Naam | Main Bridge | 6.04698 | 29.95107 | 483 | 0 | 0 | ++ |
| 6/10/23 | Naam | Kpagulobe | 6.05217 | 29.94271 | 496 | +++ | +++ | ++ |
| 7/10/23 | Naam | Ninacross | 6.0429 | 29.95056 | 485 | 0 | 0 | 0 |
| **YEI RIVER SURVEYS-2024** | | | | | | | | |
| 1/03/24 | Yei | Ifo | 5.38495 | 30.31105 | 553 | 0 | 0 | 0 |
| 1/03/24 | Yei | Wulikori-1 | 5.39207 | 30.31073 | 533 | +++ | ++ | ++ |
| 1/03/24 | Yei | Wulikori 2 | 5.39374 | 30.31073 | 555 | +++ | ++ | ++ |
| 1/03/24 | Yei | Dongoro village | 5.36796 | 30.31841 | 536 | +++ | + | ++ |
| 2/03/24 | YeI | Borro falls | 5.49379 | 30.32886 | 519 | ++ | + | 0 |
| 3/03/24 | Yei | Tawa/Amadi | 5.51481 | 30.33211 | 499 | +++ | +++ | + |
| 4/03/24 | Yei | Peke-1, Kulundu | 5.62226 | 30.30915 | 502 | +++ | +++ | 0 |
| 4/03/24 | Yei | Peke-2 | 5.61667 | 30.31062 | 502 | +++ | ++ | 0 |
| 6/03/24 | Yei | Bari falls | 5.22828 | 30.39508 | 618 | + | 0 | 0 |
| 8/03/24 | Yei | Walaka rapids | 4.75428 | 30.48662 | 663 | +++ | +++ | 0 |

Immature stages: 0 –no larvae or pupae; + up to 10; ++ 11–50; +++ >50; Adult fly: 0- no fly caught,; + up to 5 flies; ++ 6–20.

**Table 2. Baseline daily biting rate of *Simulium damnosum* s.l. along River Naam in Mvolo County (October 2023) and River Yei in Mundri West County (March 2024).**

| River | Catching site | Person days worked | Number of flies caught | Bites/Person/Day (BI/P/D) |
|---|---|---|---|---|
| Naam | Damilara | 7 | 283 | 40.4 |
| | Bridge | 7 | 270 | 38.6 |
| | Kpatiborokgba | 7 | 207 | 29.6 |
| **TOTAL River Naam** | | **21** | **760** | **36.1** |
| Yei | Wulikori | 9 | 122 | 13.6 |
| | Dongoro | 9 | 70 | 7.8 |
| | Borro | 7 | 353 | 50.4 |
| | Tawa | 7 | 264 | 37.7 |
| **TOTAL River Yei** | | **32** | **804** | **25.5** |

Wulukori site, with 7.8 bites per person per day, while the highest BI/P/D was recorded at the Borro site in Amadi, reaching 50.4 bites per person perday. The overall mean BI/P/D for the four sites was 25.5 bites per person per day (Table 2).

The details of the 11-hour diurnal catch along the River Yei are shown in Fig 3. Data collected between 06:00 and 17:00 suggest a bimodal activity pattern, with peak biting periods occurring in the morning (06:00–08:00) and evening (16:00–17:00). At the Dongoro and Tawa sites, morning peaks were not clearly observed and in Tawa there was no increase of biting rates in the evening.

**Slash and clear implementation and monitoring**

a) **Effectiveness of S&C along River Naam**

The *S. damnosum* fly densities pre- and post-S&C along the River Naam are shown in Fig 4. A sharp reduction in the blackfly population was observed for three consecutive months at both the intervention site and the Main Bridge control site, whereas at the Domilara control site, the reduction was sustained for only two consecutive months. MBR change

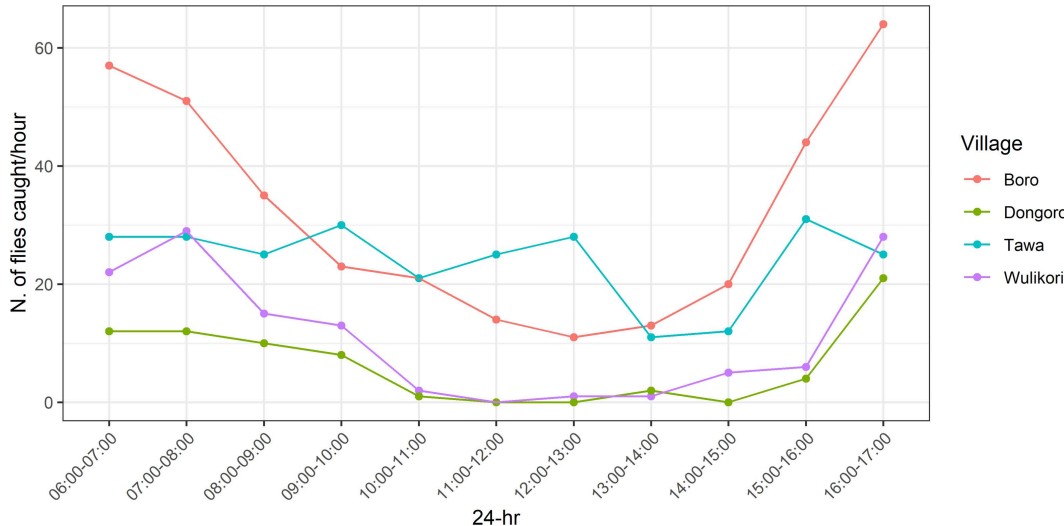

**Fig 3. Diurnal biting activity of *S. damnosum* along River Yei in Mundri West.**

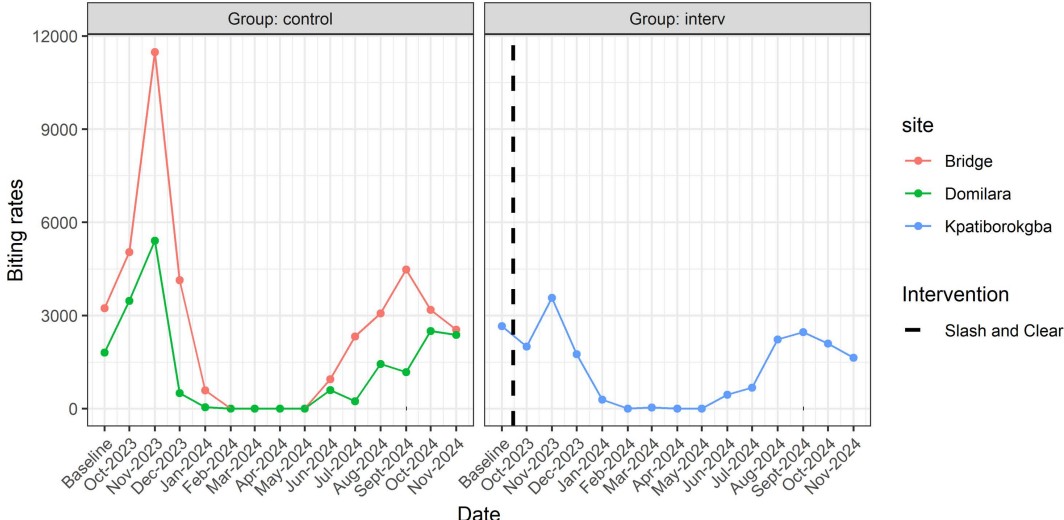

**Fig 4. Graphical illustration of the evolution of monthly biting rates along River Naam River in Mvolo County.** *Dotted vertical line represents the timing of S&C.*

from baseline to the first month of follow-up on River Naam (October 2023) was calculated as -24.5% reduction for the intervention site (Kpatiborogkba), and +69.0% increase for the control sites (mean MBR reduction for the two control sites: Damilara and Bridge). These changes are illustrated in Fig 4.

The Mann-Kendall test to assess MBR evolution on River Naam found no significant trend both at the intervention site (tau = -0.038, C) and the control sites (tau = -0.135, p-value = 0.313).

**b) Effectiveness of slash and clear along River Yei**

Along River Yei some sites were found to be favorable for S&C based on their ecological characteristics (see Fig 5).

The evolution of MBR with and without S&C along the River Yei is illustrated in Fig 6. Calculating the change in MBR between baseline and the first month of follow-up found an increase of +228.4% and +91.2% in the intervention and control sites respectively. Moreover, the Mann-Kendall test found no significant MBR trends along River Yei, both for intervention sites (tau = 0.046, p-value = 0.820) and control sites (tau = 0.163, p-value = 0.363).

**c) Generalized Additive Models:**

Regarding the Naam River in Mvolo, the log-transformed biting rates did not differ significantly between intervention and control sites; seasons (rainy vs dry) did not also show any significant effect in the fixed part of the GAM. However, the season and time interaction (te(season, time)) was highly significantly associated with log-transformed MBR (p-value<0.001); see Table 3. The estimated effective degrees of freedom (EDF) for the GAM at the Naam River was 7.89, indicating that the MBR pattern greatly varied across seasons and in a non-linear fashion. On the other hand, the time by intervention group interaction (te(time, group)) resulted in an EDF of 0 (p-value = 0.588), implying little or no time-varying effects on MBR across intervention and control sites.

Turning to Mundri, none of the fixed covariates showed any significant association with log-transformed MBR values (all p-value>0.05). Notwithstanding, the time-season interaction (te (time, season)) was highly significant (p-value<0.001) and had an estimated EDF of 3.44. This indicates that the trend over time in biting rates differed significantly between rainy and dry seasons, explaining a nonlinear seasonal time trend (Table 4). Conversely, the time by intervention group interaction (te (time, group)) was not significant (p-value = 0.204), with an EDF close to zero. This means that the intervention did

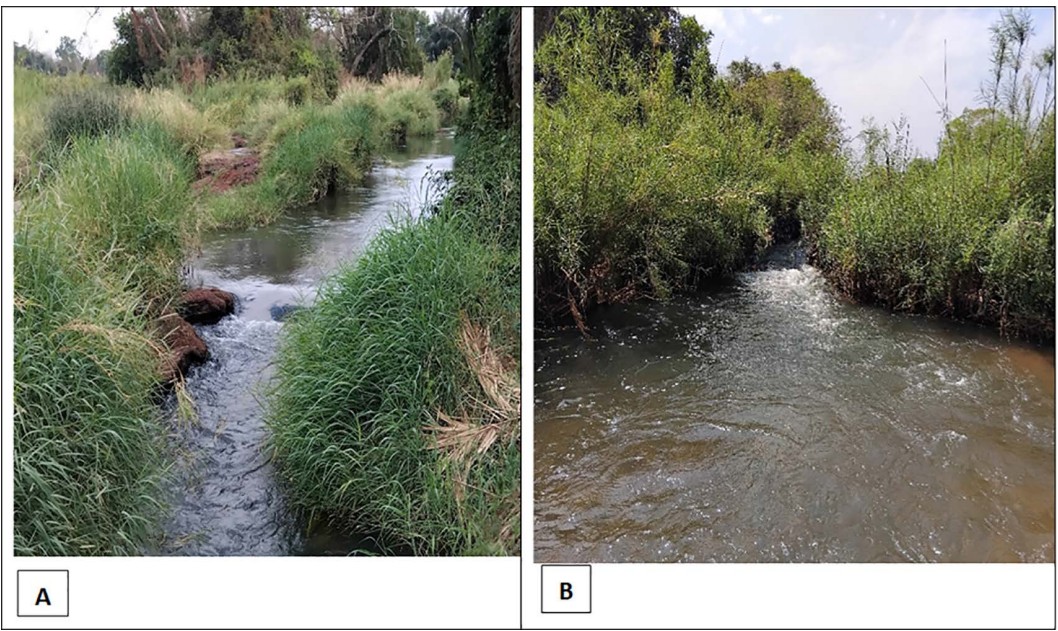

**Fig 5. Panel A, showing a stretch with trailing vegetation at Wulikori site close to Mundri Town; Panel B, Tawa site in Amadi showing one of the five sub-tributaries.**

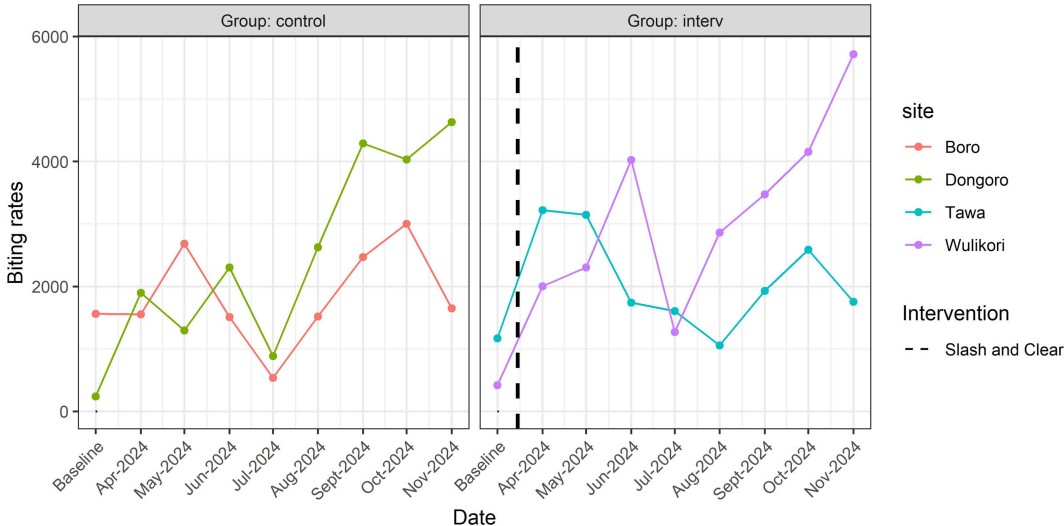

**Fig 6. Graphical illustration of the evolution of monthly biting rates along Yei River in Mundri West County, Western Equatoria, South Sudan.** *Dotted vertical line represents the timing of S&C.*

not influence the temporal pattern of biting rates – the trend over time was not distinct between intervention and control sites (Table 4).

The fitted values from both models closely match the observed values in the two rivers, with the models accounting for a high proportion of the deviance in MBR values: 87.6% for the Naam River in Mvolo (S1 Table), and 54.1% for the Yei River in Mundri (S2 Table).

PLOS Neglected Tropical Diseases

**Table 3. Principal and smooth effects from the generalized additive model (GAM) on log-Transformed monthly biting Rates in Mvolo.**

| Principal effects (fixed terms) | Estimate | Statistic | p-value |
|---|---|---|---|
| (Intercept) | 32.079 | 0.666 | 0.510 |
| Intervention vs Control | -30.774 | -0.456 | 0.651 |
| Rainy vs Dry | 0.426 | -0.639 | 0.527 |
| **Smooth terms** | **Effective degrees of Freedom (EDF)** | **F-statistic** | **p-value** |
| te (time, season*) | 7.893 | 23.777 | **<0.001** |
| te (time, group**) | 0.000 | 0.000 | 0.588 |

*season: rainy vs dry.

**group: control vs intervention.

**Table 4. Principal and smooth effects from the generalized additive model (GAM) on log-transformed monthly biting rates in Mundri.**

| Principal effects (fixed terms) | Estimate | Statistic | p-value |
|---|---|---|---|
| (Intercept) | 7.145 | 15.620 | **<0.001** |
| Intervention vs Control | 0.347 | 0.635 | 0.531 |
| Rainy vs Dry | 0.187 | 1.126 | 0.269 |
| **Smooth terms** | **Effective degrees of Freedom (EDF)** | **statistic** | **p-value** |
| te (time, season*) | 3.444 | 5.101 | **<0.001** |
| te (time, group**) | 0.005 | 0.002 | 0.204 |

*season: rainy vs dry.

**group: control vs intervention.

## Discussion

*S. damnosum* breeding was confirmed along the Yei and Naam Rivers, primarily in rapids. One distinct type of larva, characterized by pronounced tubercles and spatulate scales, appeared to be dominant across most sites. The limited data available on this larva suggest that it may be associated with the anthropophilic *S. damnosum* s.l. [16]. While the exact cytospecies could not be determined through morphological identification, it is suspected to be *S. sirbanum*, a cytospecies known to be a vector in nearby Maridi County.

Larval productivity varied across sites along the Naam and Yei Rivers. The heavy infestation (>50 larvae/pupae) observed at most sites along the River Yei are likely attributable to the river's larger size, abundant substrates, and favorable physiochemical conditions. These ecological factors, including vegetation interspersed among prominent rocks, create areas of high-velocity water flow hat support blackfly breeding. Previous studies have shown that environmental factors such as river size, discharge rate, pH, conductivity, and temperature significantly influence the productivity of a breeding site [18].

We found that breeding sites along the River Naam occured primary in rapids formed by the river crossing a chain of large rocks, as previously reported by Lewis [19]. These rocks are largely devoid of vegetation and offer limited surfaces for the immature stages of *S. damnosum t*o anchor, likely contributing to the low larval densities observed.

Our investigations into the biting patterns of sibling species of *S. damnosum* s.l. along the Naam and Yei Rivers have provided a clearer understanding of the daily biting rate and diurnal timing along the River Yei. Variations in daily biting rates were observed along both rivers, likely reflecting differences in ecological settings. Crosskey previously reported that *S. damnosum* s.l. biting levels were more stable in forested areas compared to savannah regions [20]. This is consistent with our observations along the Naam and Yei Rivers, where weak bimodal biting peaks —primarily in the morning—were noted in the savannah zones of Mvolo and Mundri West Counties. Similar but less pronounced morning biting peaks were

also observed in Tawa and Boro, two savannah sites along the River Yei. These patterns are in line with earlier reports from all sites along the River Naam [11].

Dry savannah regions such as Western Equatoria state of South Sudan typically experience higher temperatures and lower humidity levels around midday, likely contributing to the declining in blackfly biting activity during this period. In contrast, daily temperature and humidity fluctuations are less pronounced in forested areas, which may could for the more stable diurnal biting patterns observed in forest-dwelling *Simulium* species [21]. Additionally, the extreme high-temperature anomalies reported by the South Sudan Weekly Weather Forecast [15] may have influenced the DBR and diurnal biting patterns along the River Yei. The bimodal peaks in biting activity—occurring in the morning and evening—observed in our study are consistent with recent findings from Maridi County [9] and other savannah regions across Africa [22].

The performance of S&C along the Naam and Yei Rivers varied due to their ecological settings and the timing of the intervention. Considering changes that occurred immediately after S&C along the River Naam, we report a reduction in MBR at the intervention site while MBR instead increased at the control sites. This indeed suggests that upon destroying blackfly breeding grounds at the intervention site, fewer vectors were able to emerge during the subsequent blackfly reproduction cycle. The increase in MBR observed in control villages could be attributed to increased rainfalls during the month of October which, in the absence of any intervention, would result in increased river flow and more blackflies. The 24.5% reduction in biting rates is modest compared to previous observations in Maridi County, where S&C often resulted in >90% decrease in blackfly bites during the following month [22]. The lower S&C impact in Mvolo could be attributed to several factors, including the suspected breeding on rocks, as previously suggested [11]. This suggests that when there are multiple breeding sites that are not associated with vegetation, the effectiveness of the S&C intervention may be compromised. Also, the fact that S&C in Maridi was done in December (start of the dry season) could imply that MBR was already declining due to the decreasing rainfall and the 90% decrease was not solely due to the intervention. This contrasts with the current study in Mvolo, where S&C was done in October amidst high rainfall levels, fostering blackfly abundance despite the intervention.

Contrasting with the Naam River, increase in MBR was observed in all sites on the Yei River in Mundri, even more importantly at the intervention sites that benefitted from S&C. While this was an unexpected finding, it could be due to the river's large size, which likely contains several productive breeding sites that were not identified during the prospection, potentially serving as sources of blackfly reinfestation for the intervention sites. Additionally, reinfestation may have occurred from nearby tributaries within a 20 km radius, given that the River Yei has several sub-tributaries. The contamination of intervention sites by control sites in this study was attributed to the short distances between control and intervention sites. In Uganda, a country neighbouring South Sudan, *S. damnosum* s.l was reported to disperse up to a distance of 55 km from the breeding site [23]. Furthermore, seasonal influence could also explain why blackflies are on the rise in the Yei River Indeed, the baseline MBR and S&C intervention all happened in March 2024, and biting rate follow-up commenced in April 2024 coinciding with the start of the rainy season. It is therefore understandable that with the returning rains, water levels at the river increased and this favoured the rapid increase in blackfly populations in the breeding sites. Hence, the seasonal trends in blackfly abundance could have masked the effects of the S&C intervention. This line of thought is supported by previous literature clearly showing a correlation between rainfall levels and blackfly biting rates [23], and also by the GAM regression findings which uphold the significant influence of season on the observed MBR (Tables 3 and 4).

The inconsistency in the performance of S&C in large rivers is not unprecedented. A recent study along the St. John River in Bong County, Liberia, involving forest species *Simulium yahense*, observed inconsistent suppression of blackfly biting rates for only a few months [24]. Similarly, in Cameroon, along the large River Mbam, the reduction in blackfly populations attributable to S&C was only 32.9% [25]. Although there was a significant reduction in blackfly numbers in areas with persistent onchocerciasis along the Mbam valley, the performance was not considered as effective as in smaller rivers, where S&C resulted in 89–99% reductions in blackfly biting rates [17]. For the River Yei, due to the poor

performance of S&C, it may be necessary to explore alternative approaches such as utilizing canoes for prospection in deeper waters and during the S&C activities. Additionally, timing the S&C activities during the dry season, when water levels are lower and breeding sites are more accessible, may improve the effectiveness of the intervention. Repeating S&C several times during the year would also certainly help to maintain biting rates at low levels, thereby curbing onchocerciasis transmission.

Our study had several limitations, and the results should be interpreted with caution. First, there was a lack of suitable sites for S&C, particularly along the River Naam, which led the research team to use only one intervention site instead of the two originally planned. The ecological habitat of the River Naam near the Mvolo settlement, as described by Lewis, is characterized by a chain of emboldened rocks [19], which limits vegetation growth and made it difficult to identify appropriate sites for S&C. The success of S&C depends on the availability of vegetation that supports the survival of the young stages of blackflies and can be easily removed with a sharp machete. However, suspected *S. damnosum* breeding on rocks complicated the intervention efforts.

Second, seasonal variations in the study areas affected the timing of the S&C implementation. Along the River Naam, the intervention took place in October 2023 (rainy season), while along the River Yei, it occurred in April 2024 (end of dry season). These differences were due to the rainfall patterns in Mvolo and Mundri West Counties. However, the follow-up biting rates reported for both sites cover both rainy and dry seasons.

Third, no water quality measurements were conducted for the two rivers, meaning that important ecological parameters, such as temperature, pH, conductivity, and oxygen content, were not measured. This limited our ability to understand the role of these factors in the observed ecological variations.

## Conclusion

The study provided valuable data on the distribution of *S. damnosum* breeding sites, MBR, diurnal biting activity and performance of S&C along the Naam and Yei Rivers. Although our study did not demonstrate a significant decrease in the MBR patterns during the months following S&C in Mvolo and Mundri West Counties, repeated implementation of this intervention with the right timing (towards the end of the dry season and throughout the rainy season) could be effective in reducing *Simulium* biting rates in endemic communities.

## Supporting information

**S1 Table. Results of un-paired t-test statistic between intervention and control sites along Naam River in Mvolo County, Western Equatoria, South Sudan.**
(DOCX)

**S2 Table. Results of un-paired t-test statistic between intervention and control sites along Yei River in Mundri West County Western Equatoria, South Sudan.**
(DOCX)

## Acknowledgments

Appreciation is extended to the Government of South Sudan, particularly the Ministry of Health, and Amref Health Africa, South Sudan office. We are indebted to the Amref health Africa Country Manager, MR Morrish Humphrey Ojok, for his administrative support during the visit and field activities.

This work would not have been possible without the support of the Directors of Mvolo and Mundri West Health Departments. We also thank Ismael for his exceptional support in compiling the list of sites to be visited along the Yei River, liaising with local authorities, and managing the security arrangements during the second field visit. Finally, we are grateful to Mr Patrick Buyinza for assisting with the statistical analysis of our data.

## Author contributions

**Conceptualization:** Thomson Luroni Lakwo, Stephen Raimon Jada, Robert Colebunders.

**Data curation:** Thomson Luroni Lakwo.

**Formal analysis:** Thomson Luroni Lakwo, Marina Saleeb.

**Funding acquisition:** Stephen Raimon Jada, Robert Colebunders.

**Investigation:** Thomson Luroni Lakwo, Stephen Raimon Jada, Peter Alinda, Moses Tionga, Constantino Doggale Remijo Marcello, Deng Gai Dual War.

**Methodology:** Thomson Luroni Lakwo, Robert Colebunders.

**Project administration:** Stephen Raimon Jada.

**Resources:** Robert Colebunders.

**Supervision:** Thomson Luroni Lakwo, Stephen Raimon Jada, Robert Colebunders.

**Validation:** Thomson Luroni Lakwo.

**Visualization:** Thomson Luroni Lakwo, Stephen Raimon Jada, Marina Saleeb.

**Writing – original draft:** Thomson Luroni Lakwo, Robert Colebunders.

**Writing – review & editing:** Thomson Luroni Lakwo, Joseph Siewe Fodjo, Stephen Raimon Jada, Peter Alinda, Moses Tionga, Constantino Doggale Remijo Marcello, Deng Gai Dual War, Marina Saleeb, Robert Colebunders.

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
