## [Decision Letter · Decision Letter 0]

A community-based vector control intervention “Slash and Clear” implemented in two onchocerciasis-endemic foci in South Sudan

Dear Dr. Colebunders,

Thank you for submitting your manuscript to PLOS Neglected Tropical Diseases. After careful consideration, we feel that it has merit but does not fully meet PLOS Neglected Tropical Diseases's publication criteria as it currently stands. Therefore, we invite you to submit a revised version of the manuscript that addresses the points raised during the review process.

Please submit your revised manuscript within 60 days Jul 12 2025 11:59PM. If you will need more time than this to complete your revisions, please reply to this message or contact the journal office at plosntds@plos.org. Please include the following items when submitting your revised manuscript:

We look forward to receiving your revised manuscript.

Kind regards,

Adly M.M. Abd-Alla, Prof asso.

Section Editor

Adly Abd-Alla

Section Editor

Shaden Kamhawi

co-Editor-in-Chief

Paul Brindley

co-Editor-in-Chief

**Journal Requirements:**

At this stage, the following Authors/Authors require contributions: Robert Colebunders. Please ensure that the full contributions of each author are acknowledged in the "Add/Edit/Remove Authors" section of our submission form.

Potential Copyright Issues:

- Please confirm (a) that you are the photographer of Figure 5, or (b) provide written permission from the photographer to publish the photo(s) under our CC BY 4.0 license.

- Figures 1 and 2. Please (a) provide a direct link to the base layer of the map (i.e., the country or region border shape) and ensure this is also included in the figure legend; and (b) provide a link to the terms of use / license information for the base layer image or shapefile. We cannot publish proprietary or copyrighted maps (e.g. Google Maps, Mapquest) and the terms of use for your map base layer must be compatible with our CC BY 4.0 license.

5) We note that your Data Availability Statement is currently as follows: "The data reported in the submitted manuscript are provided as part of the submitted articleNo". Please confirm at this time whether or not your submission contains all raw data required to replicate the results of your study. Authors must share the “minimal data set” for their submission. PLOS defines the minimal data set to consist of the data required to replicate all study findings reported in the article, as well as related metadata and methods (https://journals.plos.org/plosone/s/data-availability#loc-minimal-data-set-definition).

- The points extracted from images for analysis..

6) Please amend your detailed Financial Disclosure statement. This is published with the article. It must therefore be completed in full sentences and contain the exact wording you wish to be published. Please ensure that the funders and grant numbers match between the Financial Disclosure field and the Funding Information tab in your submission form. Note that the funders must be provided in the same order in both places as well.

**Reviewers' Comments:**

Reviewer's Responses to Questions

**Key Review Criteria Required for Acceptance?**

**Methods:**

-Are the objectives of the study clearly articulated with a clear testable hypothesis stated?

-Is the study design appropriate to address the stated objectives?

-Is the population clearly described and appropriate for the hypothesis being tested?

-Is the sample size sufficient to ensure adequate power to address the hypothesis being tested?

-Were correct statistical analysis used to support conclusions?

-Are there concerns about ethical or regulatory requirements being met?

Reviewer #1: I believe the hypothesis is clear and that the study design appropriate to the challenging setting the authors worked in.

I am not a statistician so I would defer any opinion on the methodology to someone better qualified to judge. I am concerned that the ethical issues around human landing catches are not articulated and there is no reference to this aspect of the study. The oncho community are seeking to find alternative approaches to the issue and this needs to be referred to

Reviewer #2: • Line.111. What was the approximate distance between the study and control sites? The figures are not titled and are rather blurred, but in Figure 1, it looks as if the study site was flanked by the control sites; if this was the case can the authors comment on whether the interventions at the study site could have contaminated the findings at the control sites?

• Lines 150-151: The sentence “Prospections in the two rivers (Naam and Yei) were conducted by experienced technical staff,…” could be reworded for clarity. The word “prospected” usually applies to the search for mineral deposits. For example: The rivers Naam and Yei were surveyed by experienced technical staff,…”. Lines 153, 180, 238, 415, Table 1: “prospections” could be replaced by “surveys”.

• Lines 240-241: can the authors give a figure for the number of remaining sites along the River Naam? Table 1 suggests there were four.

Reviewer #3: THe authors need to review the indicators used to calculate the reduction rate (% exchange rate)

Reviewer #4: Yes, the objectives of the study were clearly articulated and testable. The study design was not entirely clear, if it was a case-control or before-after intervention.

Reviewer #5: The objectives of the study are not clearly articulated with a clear testable hypothesis.

A comparison is made between the baseline Monthly Biting Rate and the subsequent MBR for 14 months. It is difficult to imagine that this Slash and Clear could have an impact on the subsequent MBR during 14 months.

**Results:**

-Does the analysis presented match the analysis plan?

-Are the results clearly and completely presented?

-Are the figures (Tables, Images) of sufficient quality for clarity?

Reviewer #1: Yes analysis matches plan.

Results well presented.

Good quality figures and Tables.

Reviewer #2: • Table 2. The heading for the last column should be Daily Mean Biting Rate (DBR).

• Figure 3. Does the figure show clearly an evening rise in biting rates at Tawa (Lines 268 – 269)? This is not clear from the figure.

• Lines 305 – 307. How confident are the authors about the reliability of the log transformed data on mean biting rates on the Naam River?

Reviewer #3: See the document attached

Reviewer #4: Yes

Reviewer #5: The results are well clearly and completely presented, but these results are based on hypotheses that does not take into consideration the biology of the Simulium and breeding sites.

**Conclusions:**

-Are the conclusions supported by the data presented?

-Are the limitations of analysis clearly described?

-Do the authors discuss how these data can be helpful to advance our understanding of the topic under study?

-Is public health relevance addressed?

Reviewer #1: Conclusions supported by data presented.

Limitations described.

Authors have a firm grasp of the implications of the study

Public health as well as programmatic relevance addressed

Reviewer #2: • Lines 343, 424. Can the authors explain what is meant by emboldened rocks?

• Lines 359-361. It is not clear from this sentence whether the Tawa and Boro sites are located in savannah regions. Could the authors clarify?

• Lines 366 – 368. Can the authors provide evidence in support of the negative effect of high temperatures on black fly breeding behaviour, even in the more stable forested areas?

Reviewer #3: Yes

Reviewer #4: Yes, however, the main results which is the impact of the S&C should standout clear and undiluted.

Reviewer #5: The conclusion is supported by the data presented, but it could not be otherwise. Having one round of Slash and Clear, it is absolutely normal there is no impact on Monthly Biting Rate during a period of 14 months.

**Editorial and Data Presentation Modifications?**

Reviewer #1: See below

Reviewer #2: • The paper could be streamlined in terms of language and punctuation, for better flow of the narrative. It would have been helpful for a native English speaker to have reviewed the text before submission.

• Lines 84, 196, 451, 460: AMREF Health Africa should be Amref Health Africa. Amref is no longer an acronym after the name change of the organization.

• It is quite hard to match the figures to the text since none of the figures are titled. In addition, they appear to be rather blurred.

Reviewer #3: Minor revisions

Reviewer #4: Accepted with minor revision

Reviewer #5: (No Response)

**Summary and General Comments:**

Reviewer #1: This is an important paper in the context of the potential approaches to controlling blackflies using the Slash and Clear (S&C) method by reducing the vegetation substrate in blackfly breeding sites which has been used in other settings. I feel that the authors need to emphasise the rationale that vector control is an important adjunct to the overall objective of elimination of onchocerciasis in Africa in combination with ivermectin MDA. The effectiveness of vector control (using insecticide) in certain settings needs to be expressed (eg Bioko, Uganda foci) assuming they are isolated which the sites studied in South Sudan are not. However, the study does show that the challenges of this approach really cannot be used and/or sustained over the vast areas of oncho endemicity placing in doubt the likelihood that S&C can be used at the necessary scale. This needs to be contextualised as the feasibility of insecticide is also a major challenge for environmental, cost and sustainability reasons. I would encourage the authors to add a paragraph raising these issues. Essentially, if WHO targets for oncho elimination are to be reached on the scale required in the diverse oncho geographies then vector control, however desirable, is not likely to have a meaningful role.

Reviewer #2: This study is limited by the single slash and clear intervention at each site, and the follow up over a relatively short period of time, giving inconclusive results. The authors themselves state (P443 – 446) that “repeated implementation of this intervention with the right timing (towards the end of the dry season and throughout the rainy season) could be effective in reducing Simulium biting rates in endemic communities”. Although the paper provides some interesting information, it would be considerably strengthened if data could be collected over several interventions and over a longer period of time. In addition, the distance between control and study sites is not provided (they appear to be quite close on the maps), so it is hard to know how much the interventions could have impacted the control sites.

Reviewer #3: This paper, whose data present a trial of physical control of onchocerciasis vectors, is very important. These data allow us to further explore the contours associated with the implementation of this black fly control strategy to eliminate onchocerciasis in endemic areas. However, the data presented in this paper suggest that a single round of S&C cannot impact MBR in the medium or long-term. The success of this strategy requires repeated rounds of S&C implementation each year with the right timing to sustainably decrease black fly biting rates.

Reviewer #4: This is a great work especially as slash and clear has not been overly exploited as a vector control tool. However, I had a bit of concern especially with the study design. It was unclear to me if this was a case control study or an intervention study (before-after).

Secondly, in the abstract, I think the results and conclusion do not emphasize the purpose of the work. From my understand, the study was to assess the impact of the community-based vector control S&C. The other results such as the BR being higher in one season than the other, etc., are more secondary outcomes but they overshadowed the rationale behind the study. I think the main objective should stand out esp. in the abstract. This was however elaborated in the author summary.

Reviewer #5: This paper by Thomson Luroni Lakwo et al. is a contribution of the experimentation of the impact of the Slash and Clear strategy on the control and elimination of onchocerciasis. This strategy has shown a significant reduction in the Simulium Monthly Biting Rate where it was implemented. In this study by Thomson Luroni Lakwo et al, a unique round of Slash and clear was conducted and the impact evaluated throughout the following 14 months, with the evaluation of the percentage change relative to baseline Monthly Biting Rates (MBR). Considering the cycle of Simulium damnosum around a month and the cycle of the vegetation along the fast-flowing rivers that grows after one month or so, one wonders what the hypothesis was, in hoping the one round Slash and clear could have a significant impact on Simulium daily or monthly biting rate throughout the following 14 months. Whatever the analyses are, or the model, it is difficult to conceive that one round of Slash and clear could have impact on Simulium Monthly Biting Rate and hence an impact on onchocerciasis transmission.

The results of this study are trouble message concerning this environmentally friendly vector control strategy. Even if the reduction of the biting rate in some experiments is modest, it remains that this strategy induced in most cases a significant reduction of the biting rates, with monthly or two monthly Slash and clear. Nobody can imagine that one round of Slash and Clear can impact either the biting rate or the onchocerciasis transmission.

PLOS authors have the option to publish the peer review history of their article (what does this mean? ). If published, this will include your full peer review and any attached files.

**Do you want your identity to be public for this peer review?** For information about this choice, including consent withdrawal, please see our Privacy Policy .

Reviewer #1: No

Reviewer #2: No

Reviewer #3: **Yes: ** Philippe Bienvenu Nwane

Reviewer #4: No

Reviewer #5: No

**Figure resubmission:**

**Reproducibility:**



---

## [Decision Letter · Decision Letter 1]

Dear Dr Colebunders,

We are pleased to inform you that your manuscript 'A community-based vector control intervention “Slash and Clear” implemented in two onchocerciasis-endemic foci in South Sudan' has been provisionally accepted for publication in PLOS Neglected Tropical Diseases.

Best regards,

Adly M.M. Abd-Alla, Prof asso.

Section Editor

Adly Abd-Alla

Section Editor

Shaden Kamhawi

co-Editor-in-Chief

Paul Brindley

co-Editor-in-Chief

**Journal Requirements**
**:**

1) Thank you for including an Ethics Statement for your study. Please include:

i) A statement that formal consent was obtained (must state whether verbal/written) OR the reason consent was not obtained (e.g. anonymity). NOTE: If child participants, the statement must declare that formal consent was obtained from the parent/guardian.].

2) We have noticed that you have uploaded Supporting Information files, but you have not included a list of legends. Please add a full list of legends for your Supporting Information files after the references list.

3) Some material included in your submission may be copyrighted. According to PLOSu2019s copyright policy, authors who use figures or other material (e.g., graphics, clipart, maps) from another author or copyright holder must demonstrate or obtain permission to publish this material under the Creative Commons Attribution 4.0 International (CC BY 4.0) License used by PLOS journals. Please closely review the details of PLOSu2019s copyright requirements here: PLOS Licenses and Copyright. If you need to request permissions from a copyright holder, you may use PLOS's Copyright Content Permission form.

Potential Copyright Issues:

i) Figures 1, and 2. Please (a) provide a direct link to the base layer of the map (i.e., the country or region border shape) and ensure this is also included in the figure legend; and (b) provide a link to the terms of use / license information for the base layer image or shapefile. We cannot publish proprietary or copyrighted maps (e.g. Google Maps, Mapquest) and the terms of use for your map base layer must be compatible with our CC BY 4.0 license.

4) Please amend your detailed Financial Disclosure statement. This is published with the article. It must therefore be completed in full sentences and contain the exact wording you wish to be published.

5) Please ensure that the funders and grant numbers match between the Financial Disclosure field and the Funding Information tab in your submission form. Note that the funders must be provided in the same order in both places as well. Currently, the Financial Disclosure states there was no funding received.

Reviewer's Responses to Questions

**Key Review Criteria Required for Acceptance?**

**Methods**

-Are the objectives of the study clearly articulated with a clear testable hypothesis stated?

-Is the study design appropriate to address the stated objectives?

-Is the population clearly described and appropriate for the hypothesis being tested?

-Is the sample size sufficient to ensure adequate power to address the hypothesis being tested?

-Were correct statistical analysis used to support conclusions?

-Are there concerns about ethical or regulatory requirements being met?

Reviewer #1: -Are the objectives of the study clearly articulated with a clear testable hypothesis stated? Yes

-Is the study design appropriate to address the stated objectives? Yes

-Is the population clearly described and appropriate for the hypothesis being tested? Yes

-Is the sample size sufficient to ensure adequate power to address the hypothesis being tested? Yes

-Were correct statistical analysis used to support conclusions? Yes

-Are there concerns about ethical or regulatory requirements being met? Yes

Reviewer #4: (No Response)

**Results**

-Does the analysis presented match the analysis plan?

-Are the results clearly and completely presented?

-Are the figures (Tables, Images) of sufficient quality for clarity?

Reviewer #1: -Does the analysis presented match the analysis plan? Yes

-Are the results clearly and completely presented? Yes

-Are the figures (Tables, Images) of sufficient quality for clarity? Yes

Reviewer #4: (No Response)

**Conclusions**

-Are the conclusions supported by the data presented?

-Are the limitations of analysis clearly described?

-Do the authors discuss how these data can be helpful to advance our understanding of the topic under study?

-Is public health relevance addressed?

Reviewer #1: -Are the conclusions supported by the data presented? Yes

-Are the limitations of analysis clearly described? Yes

-Do the authors discuss how these data can be helpful to advance our understanding of the topic under study? Yes

-Is public health relevance addressed? Yes

Reviewer #4: (No Response)

**Editorial and Data Presentation Modifications?**

Reviewer #1: No

Reviewer #4: (No Response)

**Summary and General Comments**

Reviewer #1: I am satisfied by the extensive revisions the authors have undertaken following review.

Reviewer #4: (No Response)

PLOS authors have the option to publish the peer review history of their article (what does this mean? ). If published, this will include your full peer review and any attached files.

**Do you want your identity to be public for this peer review?** For information about this choice, including consent withdrawal, please see our Privacy Policy .

Reviewer #1: No

Reviewer #4: No

---

## [Editor Report · Acceptance letter]

Dear Dr Colebunders,

We are delighted to inform you that your manuscript, "A community-based vector control intervention “Slash and Clear” implemented in two onchocerciasis-endemic foci in South Sudan," has been formally accepted for publication in PLOS Neglected Tropical Diseases.

Best regards,

Shaden Kamhawi

co-Editor-in-Chief

Paul Brindley

co-Editor-in-Chief
